# Can ferric-oxyl excited states explain elongated iron-oxygen bonds in heme peroxidase catalytic intermediates?

Lewis J. Williams[1,2,3], Jos J.A.G. Kamps [2,3], Adrian M. V. Brânzanic [4,5], Maria Lehene [4], Kristoffer J. M. Lundgren [6], Ulf Ryde [6], Kuntal Chatterjee[7], Margaret D. Doyle[7], Philipp S. Simon [7], Hiroki Makita [7], Amy J. Thompson [2], Aaron S. Brewster [7], Tiankun Zhou[2], Marina Lučić[1], Michael T. Wilson[1], Pierre Aller [2], Juan Sanchez-Weatherby[2], Leland Gee [8], Sebastian Dehe [8], Sandra Mous [8], Junko Yano [7], Vittal K. Yachandra [7], Michael A. Hough[1,2,3] ✉, Allen M. Orville [2,3] ✉, Jan F. Kern [7] ✉, Radu L. Silaghi-Dumitrescu [4] ✉ & Jonathan A. R. Worrall [1] ✉

The use of X-ray structures to determine and interpret the ferryl iron-oxygen bond order in molecular oxygen-activating heme enzymes has, in the past, been controversial. This has mainly stemmed from the susceptibility of ferryl species to X-ray-induced electronic state changes. In this work we establishe using time-resolved serial femtosecond X-ray crystallography (tr-SFX) on a dye-decolourising peroxidase that the ferryl intermediate species (Compounds I and II) captured following in situ mixing of microcrystals with $H_2O_2$ have single, rather than the double bond character expected. X-ray emission validated tr-SFX data with quantum refinement, time-dependent-DFT calculations and QM/MM geometry optimizations together support the concept that the single iron-oxygen bond character is not an indication of ferryl reduction or a protonated form ($Fe^{IV}$-OH) but is instead attributed to the existence of accessible excited states possessing ferric-oxyl ($Fe^{III}–O^{·-}$) character. Such states offer insight into the nature of ferryl heme.

In the catalytic cycle of most heme enzymes that activate molecular oxygen, two common ferryl ($Fe^{IV}$-O) species sequentially form and are referred to as Compound I and Compound II[1]. Compound I is a $Fe^{IV}$=O porphyrin π-radical cation species, and Compound II is formed following the one-electron reduction of the porphyrin π-radical cation. The chemical nature of the ferryl heme along the reaction coordinate has long been a contentious issue[2]. Decades of spectroscopic and structural research have focused on determining Fe−O bond lengths of

the ferryl species, which can be taken as a reporter of protonation state, based on bond order; a double bond $Fe^{IV}$=O species is expected to be shorter (~1.65 Å) than a single bond $Fe^{IV}$−OH (~1.85 Å).

The type of oxidative chemistries in which Compounds I and II engage, e.g., selective C−H bond activation and substrate hydroxylation (cytochrome P450s family) versus sequential one-electron oxidations of organic substrates (peroxidase family), is thought to be largely dependent on the acid/base properties of the $Fe^{IV}$-oxo unit,

[1]School of Life Sciences, University of Essex, Colchester, UK. [2]Diamond Light Source Ltd, Harwell Science and Innovation Campus, Didcot, UK. [3]Research Complex at Harwell, Harwell Science and Innovation Campus, Didcot, UK. [4]Faculty of Chemistry and Chemical Engineering and INSPIRE Platform InfoBio-Nano4Health & Biomedical Imaging, Babeș-Bolyai University, Cluj-Napoca, Romania. [5]Raluca Ripan" Institute for Research in Chemistry, Babeș-Bolyai University, Fântânele, Romania. [6]Division of Computational Chemistry, Lund University, Chemical Centre, Sweden. [7]Molecular Biophysics and Integrated Bioimaging Division, Lawrence Berkeley National Laboratory, Berkeley, CA, USA. [8]Linac Coherent Light Source, SLAC National Accelerator Laboratory, Menlo Park, CA, USA. ✉e-mail: Michael.Hough@diamond.ac.uk; allen.orville@diamond.ac.uk; jfkern@lbl.gov; radu.silaghi@ubbcluj.ro; jworrall@essex.ac.uk

which is tuned through the electron donating ability of the proximal heme ligand (i.e., His *vs* Cys)[3,4]. In cytochrome P450s, the strong electron-donating ability of the Cys thiolate heme ligand creates a basic ferryl species, which promotes C–H bond cleavage of a bound substrate, through the formation of an $Fe^{IV}$–OH Compound II species[3]. In contrast, His-heme ligation (peroxidases) creates an electrophilic $Fe^{IV}$=O species, which disfavors protonation and results in Compound I and II participating in two sequential one-electron reactions with a substrate[1,2]. Thus, there is strong reason to predict that both Compound I and II in peroxidases are $Fe^{IV}$=O species.

The seminal work by Hajdu and co-workers that used X-ray dose to drive formation of $Fe^{IV}$=O species in cryo-cooled crystals of horse radish peroxidase, coupled with non-synchronous *in crystallo* absorption spectroscopy[5], inspired a generation of crystallographers to investigate $Fe^{IV}$–oxygen generated species using similar coupled approaches. Whilst there have been notable successes[6], the lingering questions of $Fe^{IV}$=O purity and the effects on Fe–O bond lengths of solvated photoelectrons produced from the X-ray induced photolysis of water molecules have confounded the issue within the field. Raven and Moody pursued the application of cryo-neutron crystallography to investigate the structures of cryo-trapped $Fe^{IV}$=O intermediates prepared from two different peroxidases (yeast cytochrome *c* peroxidase (C*c*P) and ascorbate peroxidase (APX)) with the intention of eliminating photo-induced alterations of the ferryl state and to observe where protons 'end up' following the heterolysis of the O–O bond[7,8]. Despite eliminating the potential artifacts from photoreduction, further controversy arose from the assignment of Compound II in APX as an $Fe^{IV}$–OH species (bond length 1.88 Å)[8]. This observation would imply that a His ligated heme can elevate its $Fe^{IV}$=O group basicity to enable facile protonation, a feature currently recognized only for the more electron-donating Cys ligated heme enzymes (e.g., cytochrome P450s)[3,4]. More recently, $Fe^{IV}$=O species have been investigated using femtosecond X-ray free electron lasers (XFELs)[9,10], where each diffraction pattern is collected faster (typically 10–50 fs) than the time required for the manifestation of X-ray induced chemical changes to the heme to occur, a principle termed 'diffraction before destruction'. SF-ROX (serial femtosecond rotational crystallography) yielded a shorter (1.7 ± 0.07 Å), Fe–O bond length for the cryo-trapped Compound I species of C*c*P, consistent with an unprotonated form[9]. However, SF-ROX gave longer cryo-trapped Compound II Fe–O bond lengths of 1.87 ± 0.046 Å and 1.76 ± 0.013 Å for APX and C*c*P, respectively[10]. The SF-ROX APX Compound II Fe–O bond length was consistent with the earlier reported cryo-trapped neutron structure[8], further implying, based on bond order, the presence of a $Fe^{IV}$–OH species[8].

Room temperature serial femtosecond crystallography (SFX) using XFELs involves delivering thousands of protein microcrystals sequentially to the XFEL beam and collecting a single diffraction image of each microcrystal. This offers the possibility to capture intermediates at room temperature following reaction initiation *in crystallo*, thus avoiding the need for cryo-trapping intermediates and therefore avoiding any risk of cryo-cooling artifacts. In addition, it is possible to capture transition metal X-ray emission spectra (XES) from the same X-ray pulse that yields all the diffraction data[11–13]. This synchronicity offers the great advantage that structural data is fully correlated with spectroscopic data, allowing for monitoring of electronic state changes of the metal ion during the experiment.

Based on the advantages of SFX highlighted above, we set out to capture the formation and evolution of ferryl heme species (Compounds I and II) in the A-type dye-decolourising heme peroxidase (DyP) DtpAa, from *S. lividans*[14]. We hypothesized that we would capture pristine $Fe^{IV}$=O catalytic states in a time-resolved manner. The solution kinetics of DtpAa have been well characterized with the addition of $H_2O_2$ to the ferric state, leading to the formation of Compound I, which then undergoes a one-electron reduction to Compound II through an autoreduction process by the removal of an electron from a Tyr residue[15,16]. We use the Drop-on-Tape (DoT) system[13,17], which allows on-demand drop-on-drop mixing of microcrystal slurries with $H_2O_2$, enabling time-resolved SFX (tr-SFX) to be carried out over the ms-s time scales at room temperature[18,19]. We also collect XES from the same XFEL pulse used to determine the tr-SFX structures, allowing for the heme electronic state to be unambiguously monitored along the reaction coordinate. We discover that the $Fe^{IV}$–O bond length does not increase between Compound I and II, but is on average 1.83 Å and therefore not in the range expected for a double bond ferryl-oxo ($Fe^{IV}$=O) species. To understand why this might be, we employ time-dependent density functional theory (TD-DFT) and excited state geometry calculations, to reveal the existence of ferric-oxyl ($Fe^{III}$–O·$^-$) electronic states. Herein, we present a case where the excitation into antibonding orbitals of the Fe-O unit, resulting in ferric-oxyl character, accounts for the 'long' Fe-O bond observed crystallographically.

## Results

### Increasing the lifetime of Compound I in DtpAa

To make Compound I more accessible for time-resolved crystallography, we constructed the Y389F variant of DtpAa. This residue is part of an aromatic dyad motif on the proximal side of the heme, which, from our previous work, was demonstrated to have a strong controlling influence on the decay of Compound I to II[16,20]. Fig. 1 compares the time-resolved solution electronic absorbance spectra of the Y389F variant with those of the wild-type (WT) enzyme following stochiometric addition of $H_2O_2$. The initial WT spectrum captured bears the hallmarks of a mixed Compound I/II species, which over time decays to a pristine Compound II species. In the case of the Y389F variant, the first spectrum captured (~60 s) has a Soret maximum at 401 nm, which is now consistent with a higher population of Compound I (an $Fe^{IV}$=O species carrying a π-porphyrin radical), relative to WT (Fig. 1)[15,21]. Notably, the Compound I to II transition is slower for the Y389F variant compared to the WT, with Compound II for the latter formed within 10 min ($t_{1/2}$ ~2.5 min), compared to 30 min ($t_{1/2}$ ~12 min) for the Y389F variant. These time profiles were corroborated by rapid mixing stopped-flow absorbance kinetics (Fig. S1), which demonstrated that Compound I remained fully populated as the sole species for up to 30 s in solution (Fig. S1).

### The SFX structure of the ferric Y389F variant

A room temperature SFX structure of the ferric Y389F variant was determined using the DoT system without the addition of $H_2O_2$ (*t* = 0). The structure was refined to 1.48 Å resolution (Table S1) and revealed a homodimer assembly in the crystallographic asymmetric unit (Fig. 2), which is the functional biological state of DtpAa[22]. The $H_2O$ structure on the distal side of the chain A heme (heme A) differs from that of the heme in chain B (heme B) of the homodimer (Fig. 2B), with a further difference between chains A and B being a positional difference in a nearby loop region (Fig. 2A). Heme A displays a well-defined spherical shaped electron density feature in both the $2mF_o$-$DF_c$ and $mF_o$-$DF_c$ maps, directly above the distal face of the heme-Fe, into which a $H_2O$ molecule (wat1) refined with full occupancy was modeled with an $Fe^{III}$–$H_2O$ distance of 2.38 Å (Table 1). This $H_2O$ molecule (wat1) further engages in an extensive distal pocket H-bond network (Fig. 2B). In contrast, for heme B, the electron density in both the $2mF_o$-$DF_c$ and $mF_o$-$DF_c$ maps above the heme-Fe distal face is 'smeared' and is best refined with a $H_2O$ molecule occupying two spatial positions modeled with 0.55 and 0.45 occupancy (Fig. 2B), with a closest distance to the heme-Fe of 2.25 Å (Table 1). This suggests the heme B distal $H_2O$ environment is more dynamic than for heme A. A plausible link to increased dynamics at heme B is the positional difference within the arrangement of residues 219–228 which are part of an extended loop region lying peripheral to each heme in the homodimer (Fig. 2A). This

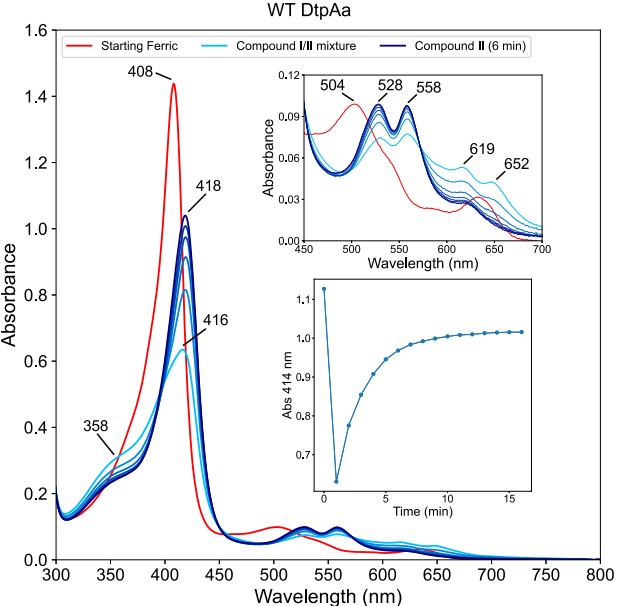

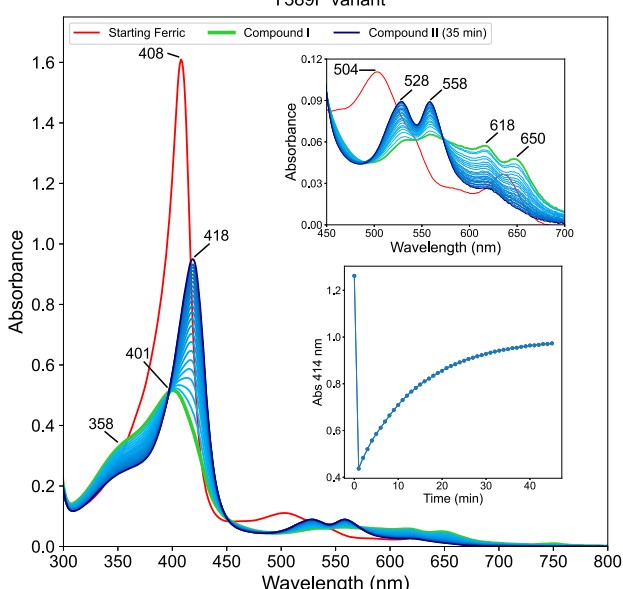

**Fig. 1 | Solution electronic absorption spectra of the WT and Y389F DtpAa variant at pH 7.0.** The starting ferric heme spectrum (red) disappears following the addition of 1 molar equivalent of $H_2O_2$ to form, within seconds, a mixed Compound I/II spectrum (light blue WT), or a Compound I spectrum (green Y389F), which then decays over time to a Compound II spectrum (dark blue). *Insets* provide a close-up of the Q-band regions (top), and the difference in Soret peak wavelengths (bottom) between Compound I and II over time. Peak maxima wavelengths for each species are indicated. Source data are provided as a Source Data file.

region is not involved in any crystal contacts, allowing for residues 219–228 to adopt either an 'in' position, which acts to shield the distal face of the heme and is predominant in chain A, or an 'out' position which greatly increases solvent exposure to the distal heme face (Fig. 2A). We note that in chain B, a stretch of disordered electron density is visible, which superposes spatially with the 'in' position occupied by the residues in chain A. This would imply that an equilibrium can exist between the 'out' and 'in' positions (Fig. S2) and may influence the bulk solvent exposure and access to the distal face of heme B.

### Time-resolved SFX structures of the Y389F variant captured by mixing with $H_2O_2$

tr-SFX data were collected using the DoT system at time-points of 0.5, 1, and 5 s, following dispensing of $H_2O_2$ onto droplets containing microcrystals of the Y389F variant. All processed tr-SFX data sets (Table S1) were initially refined against the SFX ferric structure with the distal heme coordinated $H_2O$ molecule (wat1) removed in both chains A and B. Electron density maps at each DoT time-point, revealed changes around the distal heme $H_2O$ network compared to the starting, ferric structure, that were most consistent with heme B reacting faster with $H_2O_2$ than heme A (Figs. 2 and 3). In heme B, we modeled an O atom into the $mF_o$-$DF_c$ density that forms directly above the heme-Fe in all time-point structures (Fig. 3). Subsequent rounds of refinement yielded Fe–O distances reported in Table 1. We note that the bond lengths lie somewhere between those expected for an $Fe^{IV}$=O and $Fe^{IV}$–OH species, with minimal variation in bond length. Bond-length errors representing an upper bound of uncertainty related to the resolution of the structure are also reported in Table 1. The position of the loop containing residues 219–228, which in chain B of the SFX ferric structure predominantly occupies the 'out' position, is now modeled in all the tr-SFX structures in the 'in' position, with occupancies of 0.8–0.9, suggesting that following reaction with $H_2O_2$, the highly oxidizing $Fe^{IV}$ heme becomes more protected from solvent.

For heme A, we refined the $2mF_o$-$DF_c$ electron density feature above the distal face of the heme at each time point with a full

occupancy $H_2O$ molecule. Bond lengths and associated coordinate errors across the three tr-SFX structures are reported in Table 1, showing that bond distance decreased over time. This suggests that chemistry in the microcrystals is occurring at the chain A heme and likely indicates that the observed electron density represents an average of the Compound I and starting ferric forms. An explanation for the slower reaction kinetics with $H_2O_2$ observed in the crystals for heme A may be linked to crystal lattice effects or the loop position of residues 219–228. The main crystal-traversing solvent channel, which provides access to the heme in chains A and B, was calculated using the software *LifeSoaks*[23] and is determined to have a bottleneck radius of 5.34 Å (Fig. S3A), which is larger than the projected radius of a $H_2O_2$ molecule and consequently would not hinder diffusion within the crystal to either heme pocket. Furthermore, the loop 'in' or 'out' position inflicts no influence on the calculated crystal solvent channel, and therefore, the difference in heme reaction rate does not appear to be due to a crystal lattice effect. At the molecular level, the loop position does have an influence on heme solvent accessibility (Fig. S3B). Using *Geomine*, a heme pocket volume of 879 Å³ for heme B (loop 'out') was calculated, significantly greater than the value of 734 Å³ calculated for heme A (loop 'in')[24,25]. In the 0.5 s SFX structure, where the loop adopts the 'in' position at both hemes, the computed heme pocket volume is 729 Å³ and 732 Å³, for chains A and B, respectively. Thus, in the crystals, the loop position can account for accessibility of $H_2O_2$ to the heme and govern reaction times.

To address whether the chain A heme fully reacts with $H_2O_2$ and to assess if further changes occur in the chain B heme over the time scale in which Compound II is likely to be highly populated based on solution spectroscopy studies (Figs. 1 and S1), microcrystals were incubated with $H_2O_2$ to a final concentration of 10 mM. SFX data were collected between 20 min and 50 min post-soak, corresponding to the time taken to deliver the full volume of the $H_2O_2$ incubated microcrystal slurry into the XFEL beam. The refined SFX structure determined to 1.9 Å resolution (Table S1) revealed $2mF_o$-$DF_c$ and $mF_o$-$DF_c$ electron density map features on the distal side of both hemes into which an O atom was modeled and refined at full occupancy to give Fe–O distances of 1.81–1.83 Å (Table 1). These bond lengths are

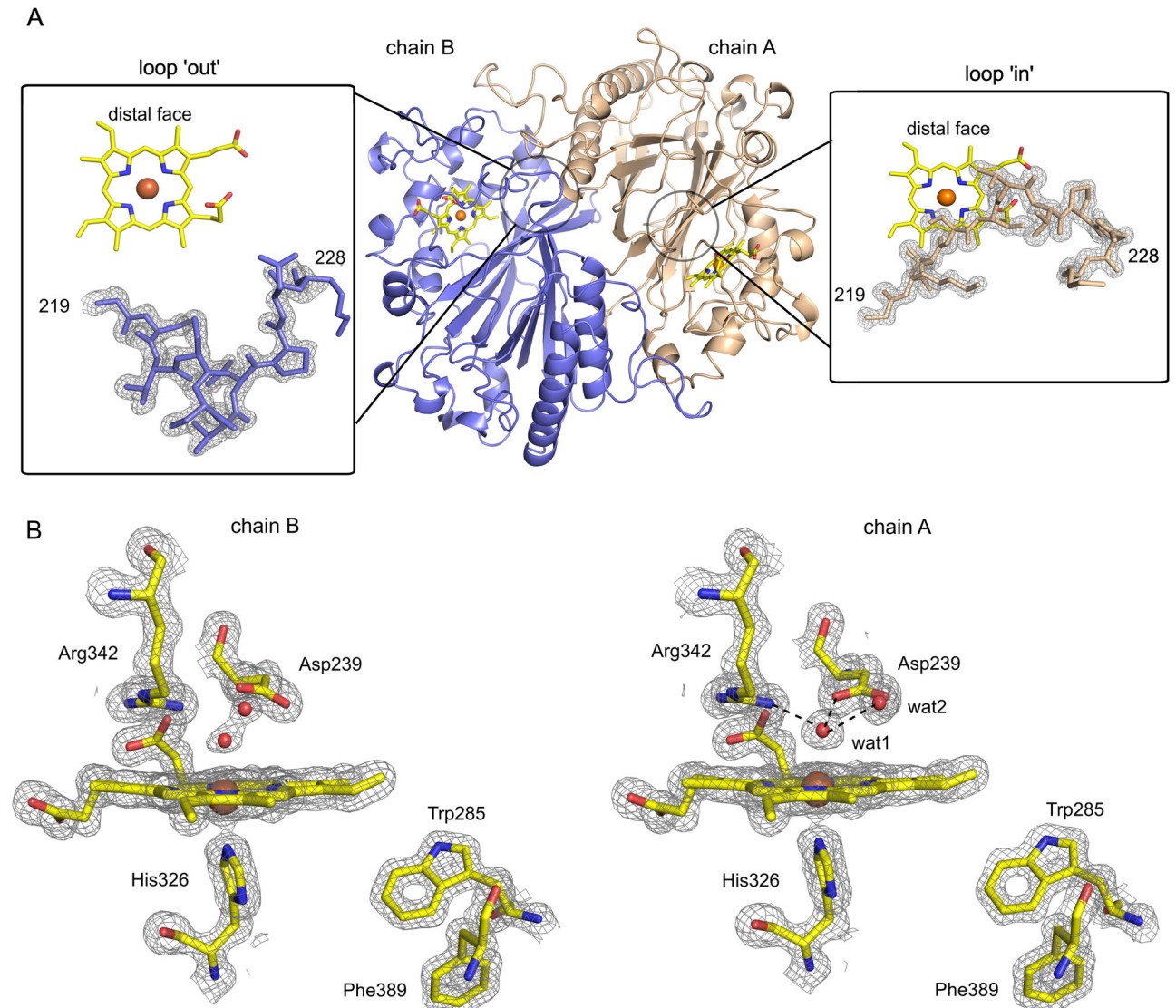

**Fig. 2 | SFX structure of the ferric Y389F DtpAa variant. A** Cartoon representation of the functional homodimer. The location of the residues (219–228) in a loop that undergoes a dynamic structural rearrangement to create an 'in' or 'out' arrangement is indicated by an open circle. The extent to which the two positions affect the exposure of the distal heme face is shown within the boxes. The arrangement of residues 219–228 in chains A and B is depicted in sticks, and the 2mF$_o$-DF$_c$ electron density maps (gray) are shown as mesh and contoured at 1.5 σ and 1.0 σ for chains A and B, respectively. For chain A, residues 219–228 refine with full occupancy, whereas in chain B, the residues refine with 0.85–0.9 occupancy. **B** The ferric heme site is shown in stick representation and the 2mF$_o$-DF$_c$ electron density maps in gray mesh, contoured at 1.5 σ. In chain A, the well-defined H-bond network of the heme Fe-coordinating H$_2$O molecule is depicted with dashed lines.

**Table 1 | Bond length and B-factors for the heme Fe and coordinated O atom in the refined tr-SFX structures of DtpA Y389F variant**

| DtpAa structure and resolution (Å) | Chain A heme | | | | Chain B heme | | | |
|---|---|---|---|---|---|---|---|---|
| | Fe–O distance (Å) | Fe–O (σ) | B-factor Fe (Å²) | B-factor O (Å²) | Fe–O distance (Å) | Fe–O (σ) | B-factor Fe (Å²) | B-factor O (Å²) |
| *ferric resting state (1.48)* | 2.38 (wat) | 0.06 | 10.2 | 12.0 | 2.25 (wat) | 0.06 | 10.6 | 12.1/12.0 |
| *t = 0.5 s (1.53)* | 2.21 (wat) | 0.07 | 10.0 | 13.5 | 1.80 | 0.07 | 10.6 | 14.7 |
| *t = 1 s (1.53)* | 2.07 (wat) | 0.07 | 11.5 | 16.0 | 1.85 | 0.07 | 11.4 | 12.8 |
| *t = 5 s (1.80)* | 2.02 (wat) | 0.12 | 12.9 | 17.5 | 1.82 | 0.11 | 13.5 | 15.0 |
| *t > 20 min (1.90)* | 1.83 | 0.14 | 13.8 | 14.3 | 1.81 | 0.15 | 16.1 | 18.8 |

Bond-length errors (σ) were computed using atomic position errors calculated by the diffraction precision indicator (DPI)[54,55]. The atomic position uncertainties for Fe and O are reported in Table S2.

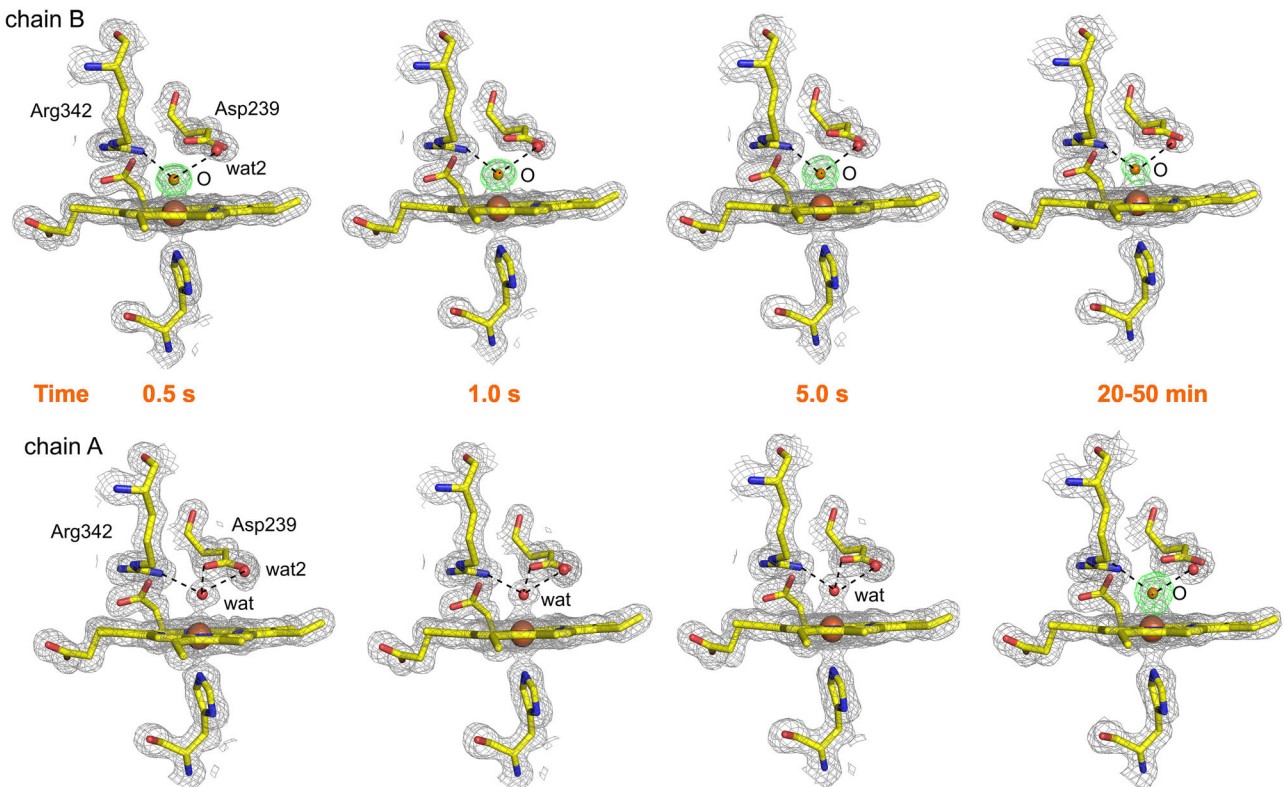

**Fig. 3 | tr-SFX structures of the chain A and B heme sites.** $2mF_o\text{-}DF_c$ electron density maps are shown in gray mesh and contoured at 1.5 σ. For the chain B heme, the modeled Fe-O atom is colored orange, and the $mF_o\text{-}DF_c$ omit map (green) calculated using Polder[83] is contoured at 5 σ. For the chain A heme, the modeled Fe-H$_2$O (wat) is shown as a red sphere across the early time series, with the 20–50 min structure modeled with an O atom and the $mF_o\text{-}DF_c$ omit map calculated using Polder[83] (green) contoured at 5 σ. H-bond interactions from the protein and wat2 molecule to the Fe-O and Fe-wat are illustrated with dashed lines.

identical to the earlier time points for heme B (Table 1) and confirm that heme A fully reacts in the crystalline state to form an Fe-O species.

### Time-resolved XES of the Y389 variant captured by mixing with H$_2$O$_2$

Whilst the distal heme electron density features of the tr-SFX structures described above bear the hallmarks of an Fe-O heme species, explicit spectroscopic verification of the heme electronic state across the DoT time series greatly strengthens the structural interpretations. We therefore collected Fe Kα (radiative 2p → 1s decay) XES data from the same X-ray pulse from which the time-resolved diffraction data were recorded. Kα XES is chemically sensitive to changes in the 2p-3d electron interactions, leading to spectral perturbations in the Kα$_1$ and Kα$_2$ bands (position, shape, and or peak height ratio), and as such, these changes are correlated with changes in the number of unpaired 3d electrons and thus spin state of the metal ion[26,27]. The normalized Fe Kα spectra for the Y389F variant in the ferric state ($t = 0$) and the subsequent time series is shown in Fig. 4A, along with the respective difference spectra (Fig. 4B). For the Kα$_1$ band, a positional shift towards lower energy (eV) and a decrease in intensity of the Kα$_2$ band following reaction with H$_2$O$_2$ are observed (Fig. 4A). By plotting the full width at half maximum (FWHM) of the Fe Kα$_1$ peak at each point of the time series, the trend reported in Fig. 4C is observed. Decreasing Kα$_1$ FWHM values have been reported for other high-spin Fe enzyme systems that undergo a spin state change accompanying an increase in oxidation state number[28]. Therefore, the Kα XES data may be interpreted with a heme spin state change, from $S = 5/2$ to $S = 1$ on reaction with H$_2$O$_2$. Figure 4C also illustrates that at 0.5 s, the ΔFWHM (-0.3 eV) is around 50% of the maximal change observed at >1 s. This correlates with our interpretation of the tr-SFX structural data, in that the chain B heme is reacting faster than the heme of chain A.

### Electronic absorbance spectroscopy of microcrystals

The tr-XES data imply that an electronic state change is occurring at the heme across the time series. This was further investigated by measuring spectra of microcrystal slurries in a cuvette using a conventional UV-vis spectrophotometer before and after the addition of H$_2$O$_2$. Prior to the addition of H$_2$O$_2$ the ferric species is clearly identifiable with a Soret band at ~404 nm (Fig. 5). One minute after addition of H$_2$O$_2$, a spectrum consistent with Compound I being the predominant species is evidenced by the presence of a 350 nm peak, and over 20 min the Soret band is red shifted to 418 nm and well-defined Q-bands at ~527 and 559 nm appear, signifying the existence of a Compound II species (Fig. 5). This time evolution mirrors that reported in solution (Fig. 1) and provides confidence that at the time-points between 0.5 and 5 s, the tr-SFX structures of the chain B heme can be assigned to Compound I and at 20–50 min to Compound II, with the latter also the case for the chain A heme.

### DFT calculations on ferryl model compounds

Across the tr-SFX structures, the Fe-O bond length following reaction with H$_2$O$_2$ is essentially the same, ~1.83 Å (Table 1), and therefore ~0.2 Å longer than expected for an Fe$^{IV}$=O species. We explored the possibility that our tr-SFX structures are more representative of a single Fe-O bond that is populated via an excited state. The classical molecular orbital diagram for an $S = 1$ ferryl system (Fig. 6A) describes the Fe-O interaction as comprising a σ bond and two halves of π bonds with notable covalent character (i.e., π and π* orbitals each featuring ~50% contributions from Fe and O, respectively). The molecular orbital diagram for the less common $S = 2$ ferryl (Fig. 6A) has two extra unpaired electrons compared to $S = 1$, derived from a single excitation from $d_{xy}$ into $d_{x^2-y^2}$; both these orbitals are non-bonding with respect to the Fe-O interaction, and

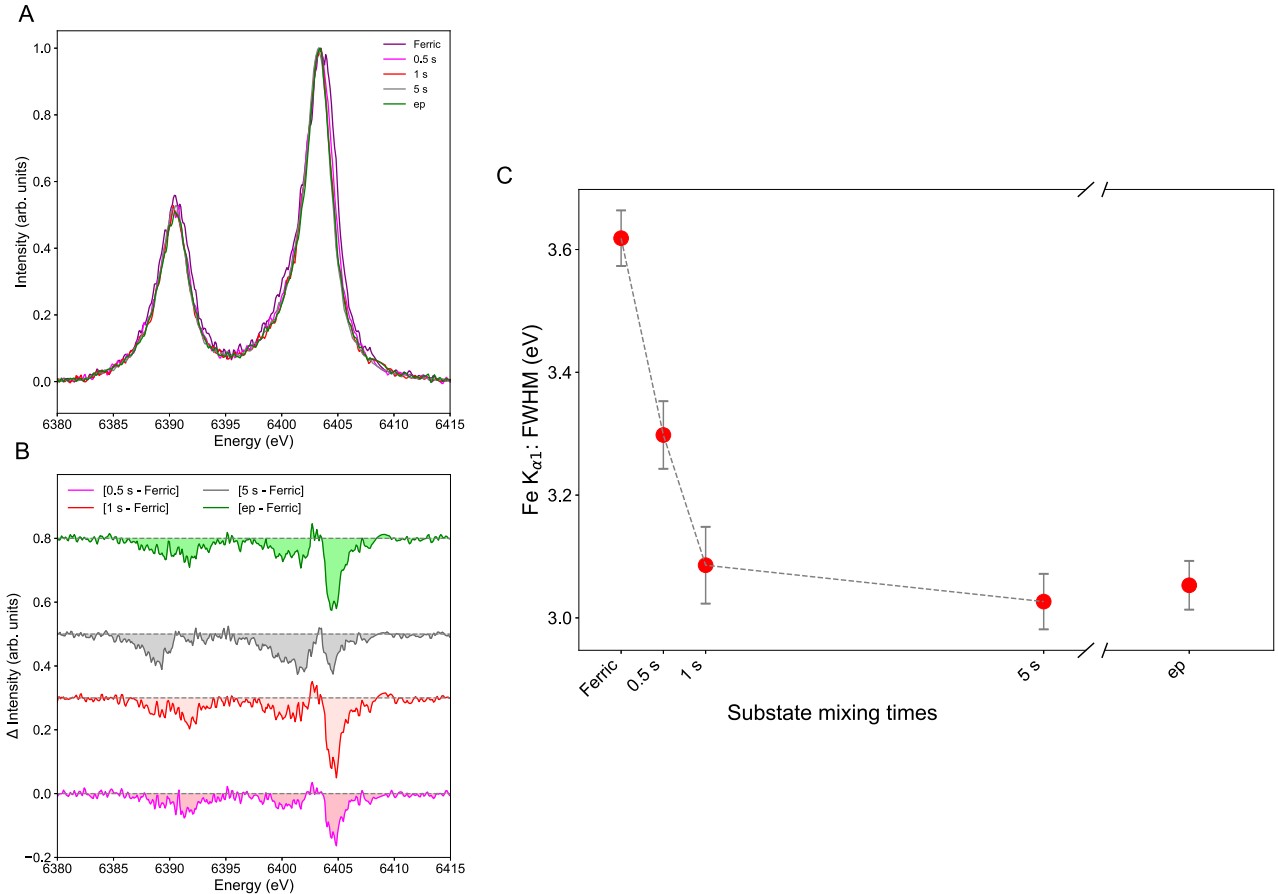

**Fig. 4 | Fe Kα tr-XES on microcrystals of the Y389F DtpAa variant.**
**A** Perturbations of the Kα$_1$ and Kα$_2$ peaks in the X-ray emission spectra relative to the ferric heme, following mixing with $H_2O_2$ over the time periods indicated and described in the main text (ep is end-point). **B** Difference spectra revealing a clear asymmetric shift in the Kα$_1$ band, as well as some weakening of intensity in the Kα$_2$ band. **C** Time-resolved plot for the changes in FWHM for the Kα$_1$ peak. The error bars for FWHM values were evaluated through a bootstrapping procedure as previously described[49]. Source data are provided as a Source Data file.

as such, the Fe−O bond order/length should not be affected. By contrast, in a putative $S = 3$ ferryl state (Fig. 6A), an electron from the π bonding orbitals ($d_{xz}+p_x$ and $d_{yz}+p_y$) is excited into the σ* antibonding orbital ($p_z + d_{z2}$), effectively reducing the bond order of the Fe−O unit from 2 to 1 and thus elongating the bond. This can be explained by six electrons in bonding orbitals −σ and π−and two electrons in the antibonding orbitals−π*−in the $S = 1$ and S = 2 states, as opposed to only five electrons in bonding orbitals counterbalanced now by three electrons in non-bonding orbitals−σ* and π*−in the $S = 3$ state (Fig. 6A). Thus, promotion of an electron from a Fe−O π (bonding) orbital into any other orbital can be expected to elongate the Fe−O bond, especially when the target orbital of the excitation is an anti-bonding σ* or π*.

DFT geometry optimizations on a heme-histidine ferryl model in the $S = 1$, 2, and 3 states described in Fig. 6A were performed, with the computed energies and Fe−O bond lengths reported in Table S3. To discriminate between the intrinsic properties of the ferryl unit and the influence of the porphyrin, a reference non-heme system ($[Fe(NH_3)_5O]^{2+}$) was also computed (Table S2). The expectation based on Fig. 6 that a $S = 3$ system would result in an increased bond length was only confirmed in the non-heme model, i.e., electrons from the π bonding orbitals are excited into the σ* orbitals (Table S3). However, for the ferryl heme model, all electronic spin states maintain a short Fe−O bond (~1.6 Å; Table S3), suggesting that a $S = 3$ excited state is not responsible for the ≈1.8 Å Fe−O distances in ferryl heme structures. These results therefore demonstrate that the porphyrin plays a role in mediating excitations with the Fe$^{IV}$−O unit.

To further explore the nature of the excitations involving the antibonding orbitals of the Fe$^{IV}$=O unit, geometry optimizations were performed on the excited states calculated using time-dependent DFT (TD-DFT) on the ground-state $S = 1$ ferryl heme model. The majority of the first 100 excited states (reported in SI) involve excitations from the molecular orbitals that contain porphyrin and Fe−O π character (orbitals β 111 and β112, red box in Fig. S4) into orbitals that display σ* character with respect to the Fe−O unit (α 121 and β 122 blue boxes in Fig. S4, and red arrow in Fig. 6B). Such transitions from bonding to antibonding orbitals would lower the Fe−O bond order and hence lengthen the bond. Conversely, we note many excitations involved transitions from the π* orbitals (α 113 and α 114 in Fig. S4, black box), which would strengthen and thus shorten the Fe−O bond (gray arrow in Fig. 6B). As expected, the optimized geometries of the first 10 excited states include Fe−O bonds of ~1.5 Å, as well as ~1.8 Å, demonstrating that excited states involving transitions into or out of antibonding orbitals can indeed lengthen or shorten the Fe−O bond distance, respectively (Fig. S5A). The ground state (GS) bond length versus energy scans for both the heme and non-heme models show energy minima at a bond length of 1.60−1.65 Å (Fig. S5B), whereas scans on higher spin states for the non-heme model showed a range of bond length minima. Notably, the first excited state of the heme model shows a minimum at ~1.85 Å and gives an unusually low energy, indicative of possible contamination with further excited states/electronic structures. This possible contamination, together with the known limits of DFT calculations in determining relative energies between spin states, electronic structures, and isomers in structures with

closely degenerate electronic states, may explain the unusually low energy of the first excited state and suggest a need for more complex calculations, i.e., QM/MM.

## QM/MM calculations on Compound II

QM/MM geometry optimization on the end-point, Compound II, structure, in the $S = 1$ GS yielded a typical ferryl-like 1.64 Å bond length (Fig. 7A). TD-DFT geometry optimization of the first excited state (i.e., the lowest-energy excited state of the $S = 1$ Compound II) remarkably led to a Fe–O bond length of 1.8–1.9 Å (Fig. 7B). The main component in the transition from the GS to the first excited state of ferryl DtpAa involves excitation from an orbital bearing Fe–O ($d_{xz}+p_x$) π character (β 243 in Fig. S6), to one with π* ($d_{xz}+p_x$) character (β 262 in Fig. S6), as

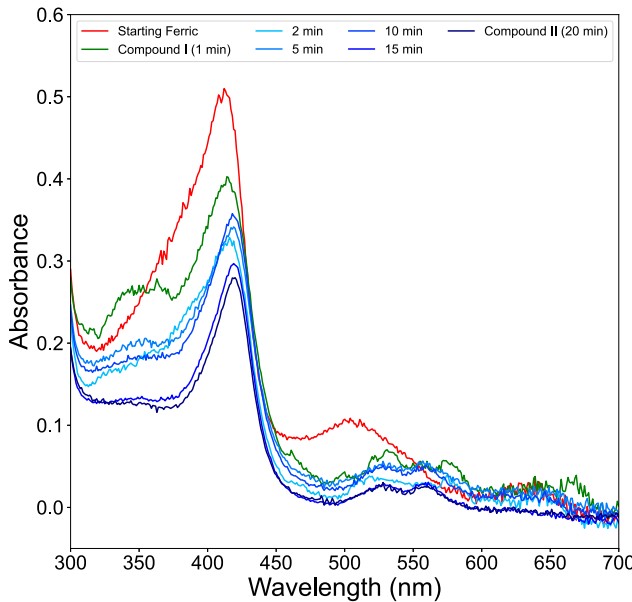

**Fig. 5 | Electronic absorption spectra of the Y389F DtpAa microcrystals.** The starting ferric state spectrum (red) is from the microcrystals grown in batch from the as-purified enzyme and used to collect SFX data at $t = 0$. Addition of $H_2O_2$ to the batch ferric microcrystals results in a spectrum with Compound I-like features (green), which over time decays to a Compound II species (purple). Source data are provided as a Source Data file.

illustrated in Fig. 7C. The Fe–O β π* orbitals in the QM/MM-optimized GS DtpAa are as expected degenerate and essentially covalent—hence with two electron holes distributed equally between the Fe and the O atoms, and an electronic structure formally described as $Fe^{IV}=O$ (Fig. 7D). In contrast, in the QM/MM-optimized first excited state (with a 1.8–1.9 Å Fe–O bond length), the two β π*orbitals are different: one is localized predominantly on Fe, and the other on O (Fig. 7D), consistent with a formal ferric-oxyl ($Fe^{III}–O^{\cdot-}$) description in the excited state. Furthermore, in this excited state, the iron-bound oxygen atom features a less negative Mulliken partial atomic charge (−0.37 vs −0.44) than in the GS, also consistent with an increased ferric-oxyl character compared to the classical $Fe^{IV}=O$. To further confirm that the geometry-optimized structure of the first excited state is not a singular artifact, QM/MM potential energy scans were performed on the 10 lowest-energy excited states of the $S = 1$ ferryl DtpAa (Fig. 7B). These 10 excited states all feature minima at Fe–O bond lengths distinctly different from the 1.6 Å seen in the GS (Fig. 7A), with several states closer to the 1.80-1.85 Å range seen across the tr-SFX structures (Fig. 7B). As was shown in the geometry optimizations on excited states of the ferryl heme model described above, these findings again reveal that changes in electronic configuration, namely excitations into and out of antibonding orbitals, with respect to the Fe–O unit, can alter the bond length, with many of these states resulting in bond lengths consistent with crystallographic observations. Based on these findings, we suggest that elongation of the $S = 1$ Fe–O bond is a result of the induction of ferric–oxyl character, weakening π-bonding interactions to give increased single bond character.

## Quantum refinement (QR) on Compound II

QR uses restraints derived from quantum chemical calculations instead of traditional restraints during structure refinement, normally applied to a small part of the overall structure[29]. We performed QR on the tr-SFX end-point structure (Compound II), focusing on the heme group in chain A and its neighboring residues. Three different hypotheses for the oxidation state of Fe and the nature of the axially coordinating ligand X, as well as the spin state, were tested, namely: $Fe^{III}–OH^-$ ($S = 1/2$), $Fe^{IV}–O^{2-}$ ($S = 1$, equivalent to the ferric-oxyl state), and $Fe^{IV}–OH^-$ ($S = 1$). QR was carried out for the GS, as well as for the first two excited states (using TD-DFT), with the three possible interpretations of Fe and X. The results are presented in Table 2 and show that a Fe–O bond length of 1.80–1.85 Å is obtained when X = $OH^-$, matching bond lengths from our tr-SFX data (Table 1). When modeling X as $O^{2-}$ (with $Fe^{IV}$ and $S = 1$), for the GS and the second lowest-energy

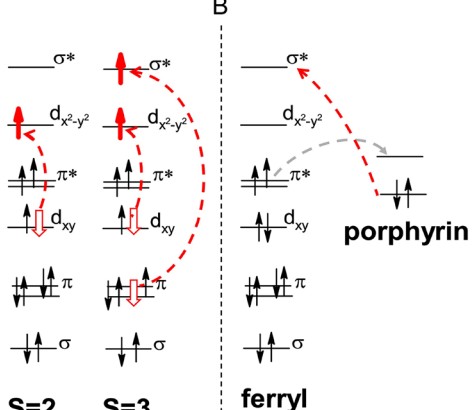

**Fig. 6 | Molecular orbital diagrams for the $Fe^{IV}$–O interaction. A** The classical $S = 1$ $Fe^{IV}$-O system, the $S = 2$ system showing excitation from the non-bonding $d_{xy}$ orbital into the non-bonding $d_{x^2-y^2}$ orbital resulting in no change to the overall bond order, and the $S = 3$ system featuring an additional excitation form the π bonding orbitals ($d_{xz}+p_x$ and $d_{yz}+p_y$) into the σ* antibonding orbital ($d_{z^2}+p_z$) resulting in an overall reduction in bond order from 2 to 1. **B** Examples of excitations between ligand (porphyrin) orbitals and antibonding orbitals, with excitations into the antibonding orbitals (red arrow) resulting in a bond order decrease and excitations out of the antibonding orbitals (gray arrow) resulting in a bond order increase.

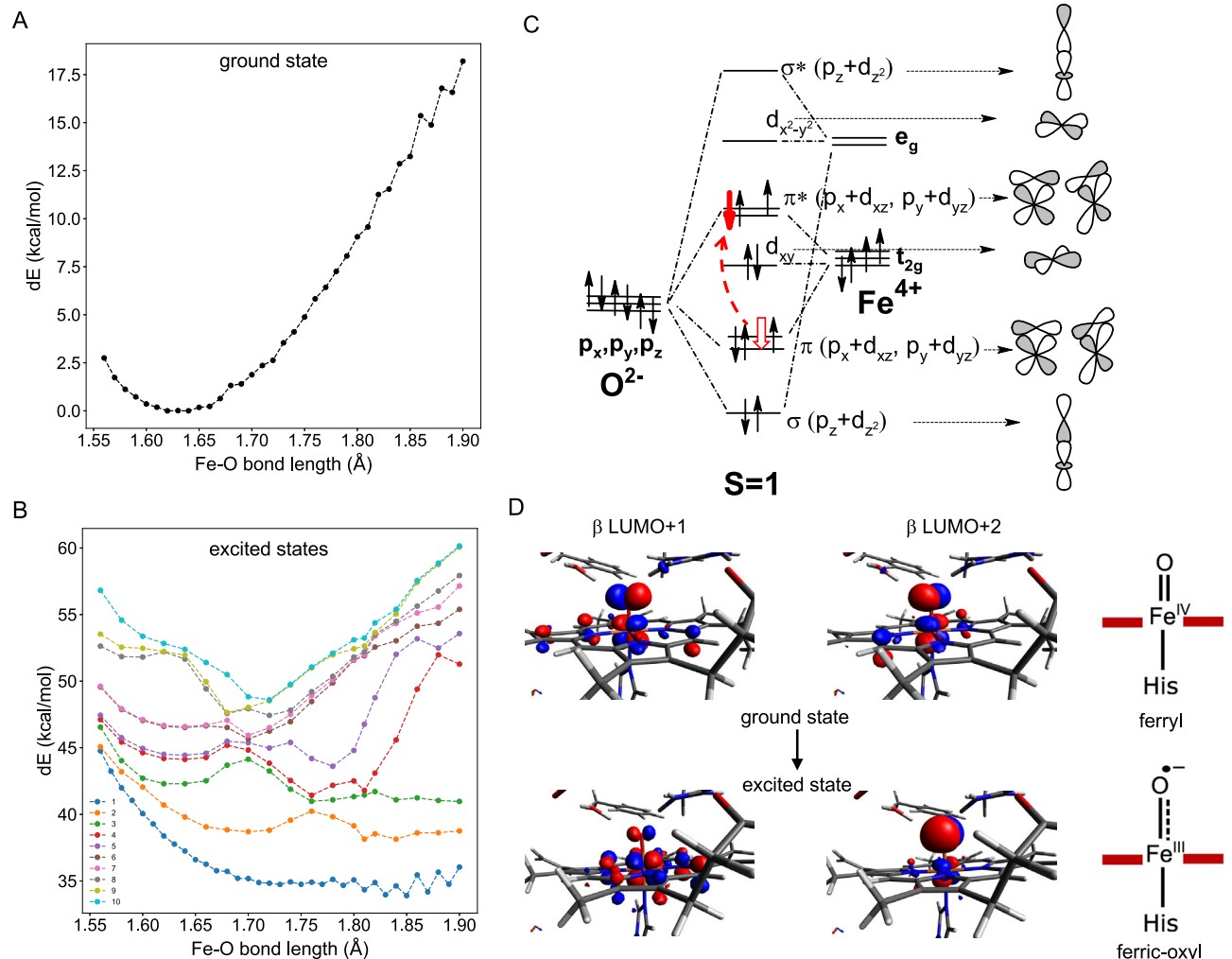

**Fig. 7 | QM/MM calculations for the GS and excited states of DtpAa Compound II.** Potential energy scans of the GS (**A**) and the 10 lowest-energy excited states (**B**) from single-point QM/MM calculations upon varying the Fe–O bond length. **C** The molecular orbital diagram of the Fe–O unit showing the primary transition (π to π*) of the S = 1 first excited state. **D** The β LUMO + 1 and β LUMO + 2 orbitals (containing π* $d_{xz}+p_x$ and $d_{yz}+p_y$ character) of the S = 1 QM/MM geometry-optimized GS versus the first (lowest-energy) excited state. The increased spin density on the iron and oxygen atoms in β LUMO + 1 and β LUMO + 2 of the first excited state, respectively, is evident by the larger lobes over these atoms. Source data are provided as a Source Data file.

**Table 2 | QR structures with a QM region consisting of heme, His326, Asp384, Asp239, Arg342, Phe363, a nearby water molecule, and the distal ligand X**

| State | RSZD(X) | Fe–O |
|---|---|---|
| $Fe^{IV}$–OH⁻, GS | 1.9 | 1.85 |
| $Fe^{IV}$–$O^{2-}$, GS | 2.5 | 1.69 |
| $Fe^{IV}$–$O^{2-}$, root 1 | 2.2 | 1.84 |
| $Fe^{IV}$–$O^{2-}$, root 2 | 2.7 | 1.69 |
| $Fe^{IV}$–OH⁻, GS | 1.1 | 1.80 |

The structures are studied either in the GS or in the first or second lowest-energy excited state (root 1 or 2).
The table shows the RSZD score of the X ligand and the Fe–$O_X$ distance in Å

excited state, short Fe–O bond distances, 1.69 Å, are obtained (Table 2). However, for the first excited state, an Fe–O bond distance of 1.84 Å is obtained with a lower real-space Z-difference (RSZD) score than for the other two X = $O^{2-}$ structures, indicating that this bond length is more favorable in the SFX structure (Table 2). Taken together, the QR gives further credence that excited states determined from TD-DFT are compatible with our crystallographic observations.

## Discussion

Time-resolved serial crystallography at ambient temperature offers the opportunity to capture transient reaction intermediates during the catalytic cycle of an enzyme[30]. Our work has comprehensively characterized the peroxidase cycle of an A-type DyP using tr-SFX, with XES and electronic absorption spectroscopy measured directly on microcrystals, providing support for the presence of fully populated Compound I and II states along the time coordinate used to obtain structures. Furthermore, we have captured multiple aspects of the dynamics involved in a heme enzyme reaction cycle, including the positional movement of a loop structure following mixing with $H_2O_2$.

The invariable Fe–O bond length across the tr-SFX series (~1.83 Å) is longer than that of a $Fe^{IV}$=O species and is consistent with a single bond. It is widely recognized that the generation of solvated electrons within crystals by the X-ray beam can result in the reduction of metal centers, which, in the case of ferryl heme, would revert to ferric or even ferrous states. Here, this is evaded with the use of SFX[31–33]. Indeed, we have recently shown that X-ray induced changes to the electronic structure of metalloproteins, demonstrated on the exact system used in this study, are negligible under standard measurement conditions for SFX[34,35]. Another factor to consider is whether the ferryl states being generated are fully populated. This has been a challenge in the

cytochrome P450 field[36], but as we demonstrate from our solution and microcrystal spectroscopies, the ferryl species generated are accessible over a ms-to-min time frame and highly populated, with the sequential formation of Compound I then Compound II, as expected. While protonation of the $Fe^{IV}=O$ unit could offer an explanation for the elongated $Fe^{IV}$-O bond in the SFX structures, especially with a cryo-trapped neutron structure of APX purportedly in the Compound II state showing evidence of protonation[8], there is, however, a strong chemical argument against protonation of Compound II in His-heme ligated peroxidases[37]. The $pK_a$ of the $Fe^{IV}$-oxo group in His-heme ligated peroxidases and globins is « 4, and therefore its electrophilic nature makes protonation highly unlikely[37]. Subsequent studies have shown the reported APX $Fe^{IV}$-OH bond length of 1.88 Å[8], to be too long for such a species and closer to a $Fe^{III}$-OH species (Table 2)[38,39]. Furthermore, in our 0.5 s time-point structure, heme B can confidently be assigned to Compound I, based on the presented spectroscopic evidence, and thus making protonation an even more unviable explanation.

DFT calculations support a picture where the two Fe=O π* orbitals, while still degenerate in energy, are not in fact covalent, but instead, one is located on the Fe while the other is on the O atom. This arrangement leads to the ferryl unit being electronically equivalent to a single bond ferric-oxyl ($Fe^{III}$-O$^{\bullet-}$) species, which previous DFT methodologies have failed to differentiate due to degeneracy of the orbitals[40]. Here we demonstrate that deviations from the GS electronic configuration and resulting population of Fe-O antibonding orbitals give Fe-O bond lengths longer than those expected for a $Fe^{IV}=O$ species, while also resulting in increased spin density on the O atom, consistent with a ferric-oxyl species (Figs. 7D and S5B). The occurrence of ferric-oxyl character in $Fe^{IV}=O$ bonds in biology has been theoretically proposed in α-ketoglutarate-dependent mononuclear nonheme iron enzymes, which utilize $O_2$ to generate $Fe^{IV}=O$ intermediates for H-abstraction of a bound substrate. DFT calculations have indicated that in the $Fe^{IV}=O$ GS, a significant proportion of the population (40%) has ferric-oxyl character with 60% 'true' $Fe^{IV}=O$ resulting in a bond length elongation of 0.14 Å[41,42].

The excited states presented in this work are low lying and accessible through absorption of lower-energy quanta, reaching down to ~700–800 nm /-1.7 eV /-38 kcal/mol required for the heme ferryl to access its first excited state according to the TD-DFT and QM/MM calculations. Furthermore, the vibrational mode computed at ~900 cm$^{-1}$ for the $Fe^{IV}=O$ unit features visible contributions from the equatorial ligands in the non-heme, but not in the heme system (Fig. S7). A possible consequence of this is that in the heme system, the π to σ* excited state would have fewer opportunities to dissipate its energy vibrationally via the remainder of the system, compared to the non-heme model, and therefore increase the chances of observing excited state character in a heme system. Therefore, we suggest that the elongated Fe-O bonds seen in our tr-SFX structures can be explained as the ferric-oxyl excited state.

In earlier SFX experiments with the hexameric DtpB (another DyP), we observed both $1.65 \pm 0.07$ Å and 1.80-1.85 Å Fe-O bond lengths across the six hemes for a long-lived Compound I state[21]. In light of the calculations reported here, we may now interpret this observation as both the ground and excited states existing within the same crystal structure, indicating that the generation of these excited states is not an artifact of X-ray radiation. A further consideration is the possibility that the electronic structure in solution can be perturbed by crystallization. Such a phenomenon has been demonstrated for oxy-hemoglobin, where a combination of factors can change the electronic configuration from $Fe^{III}$-O$_2^{\bullet-}$ in solution to $Fe^{II}$-$O_2$ in the crystal state[43]. The simulated electronic absorption spectra of DtpAa Compound II in the $Fe^{IV}=O$ and ferric-oxyl electronic configurations show high spectral similarity (Fig. S8), making discrimination between ground and excited states in our crystal spectra challenging due to signal-to-noise limitations compared to solution.

In summary, our work leads to a corollary that elongated heme $S = 1$ Fe-O bonds need not be due to protonated ferryl states but rather are interesting properties of an unprotonated ferryl. How these ferric-oxyl traits affect the energetics of electron transfer with reducing substrates requires further investigation. Future studies are now required to evaluate whether elongated ferryl Fe-O bonds are indeed a paradigm shift in understanding the chemical and catalytic nature of the $Fe^{IV}=O$ species.

## Methods

### Construction, over-expression, and purification of the DtpAa Y389F variant
A construct for the over-expression of the Y389F variant of DtpAa was prepared using the Quik Change site-directed mutagenesis procedure as described in the Supporting Information. Large-scale over-expression of the Y389F construct was carried out in *Escherichia coli* using the C43(DE3) host strain, with subsequent harvesting and purification identical to a previous report[44].

### Electronic absorption spectroscopy
A Cary 60 UV-Vis spectrophotometer (Agilent) was used to measure electronic absorbance spectra. The purified Y389F variant was exchanged into a desired buffer for experiments using a PD10 desalting column (Cytiva) and sample concentrations determined from the protein absorbance peak at 280 nm using an extinction coefficient (ε) of 46,075 M$^{-1}$ cm$^{-1}$. Working solutions of $H_2O_2$ were prepared from dilution of a stock 30 % $H_2O_2$ concentrated solution (Sigma-Aldrich) and concentrations determined ($\varepsilon_{240\ nm}$ of 43.6 M$^{-1}$ cm$^{-1}$). Following the stoichiometric addition of $H_2O_2$ to a quartz cuvette (Helma) containing the Y389F variant (10 μM), the electronic absorption spectrum between 800 and 250 nm at various time intervals up to 40 min was recorded. Rapid mixing kinetics were performed using an SX20 stopped-flow spectrophotometer (Applied Photophysics, UK) equipped with a diode-array multi-wavelength unit and thermostatted to 25 °C.

### Preparation of protein crystals
Crystallization of the ferric DtpAa Y389F variant was set up within 48 h of protein purification. Microcrystals were obtained under batch conditions by mixing in 1.5 ml microfuge tubes (Eppendorf) a 1:3 *v/v* ratio of a 10 mg/ml protein solution in 20 mM sodium phosphate, 150 mM NaCl, pH 7.0, and a precipitant solution containing 12% *v/v* PEG 3350 (Sigma-Aldrich), 100 mM HEPES, pH 7.0, to give a final volume of 400 μl. Microcrystals with dimensions of $30 \times 30$ μm grew within 48 h at 18 °C.

### SFX and XES data collection
Experiments were conducted in September 2023 as part of the L10057 proposal using the Linac Coherent Light Source (LCLS), at the MFX beamline at the SLAC National Accelerator Laboratory, Menlo Park, California. All measurements were carried out at 27 °C. XFEL data were collected with a pulse length of 30 fs, a beam energy of 9.4 keV, -1.5 mJ/pulse, and a repetition frequency of 30 Hz using a Rayonix MX340-HS detector. X-ray emission data were collected concomitant with diffraction data. The Fe K$\alpha_{1,2}$ emission signal in the energy range 6380–6415 eV was collected on a wavelength-dispersive von Hamos spectrometer with 3 cylindrically bent ($R = 250$ mm) LiNbO$_3$ (23$\bar{4}$) crystals oriented above the sample with the center of the crystals located at 80.69° with respect to the XFEL interaction point-to-detector axis[13,45,46]. The emission signal was measured using a two-dimensional ePix100 detector placed directly beneath the interaction point. The data were pedestal and gain corrected to account for differences in noise and gain of detector pixels.

## DoT mixing experiments

Drop-on-drop on-demand tape drive mixing experiments (DoT) were performed using a conveyor belt system in combination with acoustic droplet ejection (ADE1) to dispense microcrystal suspension as described previously[17,18] with minor modifications. In brief, a syringe pump holding a glass syringe (Hamilton, 1.0 mL) of a microcrystal suspension, was used to feed the ADE1 transducer through an ID fused silica capillary (250 μm), at a fixed flow rate (7 μL min⁻¹). The microcrystal suspension containing syringe, also containing a plastic bead, was rocked over an angle of 180°. The back-and-forth movement of the plastic bead, as well as rocking over 180° prevented settling of the crystals. The ADE1 transducer dispensed at a fixed frequency (30 Hz) synchronized with the LCLS master clock, creating nanolitre-sized droplets (-3.9 nL) onto the Kapton conveyor belt. For microcrystal slurries of the ferric Y389F variant and the $H_2O_2$ equilibrated sample, where no drop-on-drop on demand mixing was required, the ADE1 dispensed drops were transported at a belt speed of 300 mm/s to the X-ray interaction point, where diffraction and XES data were synchronously recorded. A $H_2O_2$ equilibrated sample was prepared by the addition of a 0.5 M $H_2O_2$ stock solution (prepared in a 3:1 precipitant:buffer(20 mM sodium phosphate, pH 7.0 150 mM NaCl) solution) to the ferric microcrystal slurry to give a final $H_2O_2$ concentration of 10 mM. Samples prepared in this way were incubated at room temperature with gentle shaking for 20 min before dispensing onto the conveyor belt using ADE1. The maximum data collection time for the equilibrated samples was 30 min, so in total, the contact time with $H_2O_2$ equilibration was 50 min. For on-demand mixing, a transistor-transistor logic (TTL) signal was passed on from the ADE1 transducer to a piezoelectric injector (PEI) through a delay generator (model DG645, Stanford Research Systems) and waveform generator (model 33612 A, Keysight Technologies). The PEI was purchased from PolyPico (miniature dispensing head), and was mounted downstream of the ADE1, above the Kapton tape, and 37.5 mm from the X-ray interaction point. The PEI was used to add bursts of droplets at 2–6 kHz of aqueous 500 mM $H_2O_2$. Dispensing frequency was dependent on Kapton tape speed (75 or 37.5 mm s⁻¹) and thus indirectly on the desired time point (0.5 or 5 s). A fast camera (Mini AX 200 from Photron) associated with a 500 mm working distance lens (ULWZ-500M from OptoSigma) observed the intersection of the droplets between the PEI and the microcrystal slurry drop from ADE1. Then, high-speed video (6000 fps) analyses acquired during data collection were used to determine the number of droplets from the PEI added to the ADE1 drop. Thus, we could estimate the added volume and hence the substrate final concentration in the resulting drop. Merging of the PEI and ADE droplets resulted in dilution of the substrate (final $[H_2O_2]$ - 203 ± 29 mM). Motorized stages were used to adjust the positioning of the PEI to align with the ADE1 droplets, as verified using two orthogonal cameras, synchronized with the ADE1 dispensing frequency. Temporal alignment of the PEI dispensed droplets with the larger ADE1 droplet was achieved through adjusting a variable delay on the waveform generator between the ADE1 and PEI, and was continuously monitored for correct alignment using a side-view camera, during data collection. To collect a time point at 5 s a second, ADE, instead of the PEI, was used for mixing (ADE2). ADE2 was fed at a fixed flow rate (3.5 μL min⁻¹) creating $H_2O_2$ droplets (100 mM in $H_2O$) half the size (1.94 nL) of ADE1 (7.0 μL min⁻¹; 3.89 nL) giving a final $H_2O_2$ concentration of 33 mM. A Kapton tape speed of 41 mm/s was used to transport the mixed drops from ADE2 to the X-ray interaction point to give a time delay between mixing and X-ray exposure of 5 s.

## Crystallographic data processing

All SFX data were processed using cctbx.xfel[47], and the detector geometry was refined using DIALS[48]. Reflection subsampling was enabled to increase the number of successfully indexed images. During ensemble refinement, the sigma_b_cutoff was set to 0.2, otherwise default parameters were used. The data resolution cut-off was defined as the point at which the correlation coefficient between half data sets ($CC_{1/2}$) decreased smoothly to 0.3. A summary of the SFX data collection statistics can be found in Table S1.

## XES data processing

Calibration of XES energies was performed by recording a Fe foil spectrum in the Fe $K\alpha_{1,2}$ energy range. Detector images were sorted as described previously[49]. For the background subtraction, a slice on either side of the region of interest (ROI) was selected. A row-by-row first-order polynomial fitting scheme was utilized for the initial two-dimensional background subtraction. To then bring the baseline of the spectra to zero, a one-dimensional background subtraction (also utilizing a first-order polynomial fitting scheme) was also deployed. The spectra were smoothed using a Savitzky-Golay filter with a window length of 5 and polynomial order of 3, and subsequently were normalized with respect to maximum intensity. In the difference spectra, resting state ferric DtpAa was considered as the reference and was subtracted from XES spectra of other time points/states. The spectral range to calculate the FWHM of the Fe $K\alpha_1$ signal was 6400–6408 eV.

## Structure determination and estimation of bond-length uncertainties

The processed SFX data were phased using the PDB entry 6i43[50]. Structure refinement was carried out in Phenix (v. 1.21-5207-000)[51], with model building between refinement cycles carried out in Coot[52]. Standard restraint libraries were used for refinement in Phenix, and the Fe–O or Fe–N(His) distances were not restrained. Errors in specific bond lengths were estimated via the diffraction precision indicator (DPI). Previously, use of the DPI to estimate coordinate error was benchmarked against the gold-standard full-matrix inversion method, with good agreement having been demonstrated[53]. A DPI value for each structure is calculated, representing the uncertainty in position for atoms with the mean refined B factor of that structure. Subsequently, each atom in the refined structure is assigned an individual value derived from the overall DPI and the B factor of the specific atom using the approach of Gurusaran and coworkers as implemented in the Online_DPI server[54,55]. The uncertainty estimates for each of the two atoms forming a bond can then be used to derive an estimate of the bond-length uncertainty as described by Helliwell (see Table S2)[56].

## TD-DFT and QM/MM calculations

For DFT calculations on small models, unless otherwise specified, geometries were optimized, and frequencies calculated using the Gaussian software package using the UB3PWP91 functional with the def2-TZV(P) or the def2-SV(P) basis sets as indicated[57]. This methodology was chosen in view of the reasonable results it provided for the spectroscopic properties (especially the TD-DFT) on other metal-macrocycle complexes (B3LYP calculations provided qualitatively similar results)[58–60]. TD-DFT calculations on small models were performed using 20 or 100 excited states, as indicated in the figure legends. For the QM/MM calculations, the coordinates of the DtpAa $H_2O_2$ soaked 'end-point' SFX structure (Compound II) were prepared in Maestro (Schrödinger Release 2025-2: Maestro, Schrödinger, LLC, New York, NY, 2025) after protonation states were assigned with the aid of PropKa[61]. The protein active site was parameterized by use of antechamber (employing AM1-BCC charges)[62] and the general amber force field (GAFF)[63] as implemented in AmberTools23[64]. The Metal Center Parameter Builder (MCPB)[65] coupled with the Seminario approach[66] was further used to prepare the computations of RESP Merz–Kollman atomic charges, which were computed at the B3LYP[67–69]/6-31 G*[70] level of theory in Gaussian16[57]. QM/MM/(MD) computations were then performed with the ORCA (version 6.0.0) software package[71,72] on the 'end-point' SFX structure. The QM and active region was set as shown in Fig S9 to incorporate the Fe–O moiety, the heme, the residual part of

the neighboring amino acids, namely Asp239 His326 (i.e., the proximal histidine), Arg342, Phe363, and the crystal water molecule WAT601 which is placed in the vicinity of the iron-bound oxo atom ("wat2" in the "Results" section). For the energy scans, the O atom was manually moved to give Fe–O bond lengths indicated in the Figures, in increments of 0.02 Å within the 1.56–1.90 Å interval, and on each geometry, single-point energies were computed at the RI[73,74]-B3PW91[67,69,75]/def2-SV(P)[76,77] level of theory. In general, the adopted DFT methodology converged higher energy wavefunctions for the 1.56 Å and 1.58 Å points, which were characterized by a higher degree of spin contamination compared to the other data points. Therefore, in all calculations concerning the 1.56 and 1.58 Å points, the converged orbitals of the 1.60 Å point were used as starting wavefunctions. This resulted in a homogeneity in spin contamination across each electronic state and ensured that all data points are described by the same electronic state. For the excited states in these QM/MM calculations, the TD-DFT approach formulated in the Tamm–Dancoff approximation[78] was employed. A window of 20 excited states was employed, unless otherwise specified.

## QR

QR, as implemented in the QRef interface[29] between the PHENIX[79] (version 1.21.2-5419) and ORCA (version 6.0.0)[71,72] software packages, was performed for several models of the heme group in chain A, starting from the non-quantum-refined structure. Two different QM regions were utilized. The smaller region consisted of the complete heme group, His326 (modeled as 5-methylimidazole), Asp384 (modeled as acetate), as well as the Fe-bound solvent molecule (called X in the following and modeled either as $OH^-$ or $O^{2-}$). The larger QM region also includes Asp239 (modeled as acetate), Arg342 (modeled as methylguanidine), Phe363 (modeled as benzene), as well as HOH-D168 (modeled as water). For the smaller model, strain energies were calculated and used to determine the relative weight to be used between the experimental data and the restraints, with the results reported in Table S4. As expected, strain energies increase, and RSZD scores decrease when increasing the weight towards the experimental data. We have previously recommended[29] to use a weight where the strain energy starts to increase, and RSZD scores start to decrease. In this respect, the results in Table S5 suggest that $w_x = 1$ is an appropriate choice. The models from QRef were evaluated in terms of RSZD scores calculated with EDSTATS[80]. For both QM regions, three different hypotheses were tested, Fe and X modeled as $OH^-$ in the low-spin state ($S = \frac{1}{2}$), $Fe^{IV}$ and X modeled as $O^{2-}$ in the intermediate-spin state ($S = 1$), and $Fe^{IV}$ with X modeled as $OH^-$ in low spin state ($S = 0$). This gives net charges of −3, −3, and −2 for both QM regions, respectively. QRef was then run using unrestricted Kohn–Sham formalism with the TPSS functional, D4 dispersion[81], and the def2-SV(P) basis set for the smaller QM region in the GS. For the larger QM region, the B3PW91 functional together with the def2-SV(P) basis set was used together with the RIJCOSX approximation (using the def2/j auxiliary basis set) and CPCM ($\varepsilon_r = 4.0$) continuum solvation calculations[82]. The latter calculations were run for both the GS and the first two excited states (nroots = 100, iroot = 1 and 2, respectively). QRef was then run for three macrocycles, where each macrocycle consisted of first coordinate refinement, in which the QM regions, as well as the remainder of the truncated residues, were allowed to move, followed by ADP refinement for the entire model. The weight between the experimental data and the restraints was manually adjusted to be either 0.1, 0.3, 1, or 3 for all models.

## Reporting summary

Further information on research design is available in the Nature Portfolio Reporting Summary linked to this article.

## Data availability

The atomic coordinates and structure factors generated in this study for the DtpAa Y389F variant time series have been deposited in the Protein Data Bank (PDB) under the accession codes; 9S5O t = 0 [https://doi.org/10.2210/pdb9S5O/pdb]; 9S5P t = 0.5 s [https://doi.org/10.2210/pdb9S5P/pdb]; 9S5Q t = 1 s [https://doi.org/10.2210/pdb9S5Q/pdb]; 9S5R t = 5 s [https://doi.org/10.2210/pdb9S5R/pdb]; 9S5S t > 20 min [https://doi.org/10.2210/pdb9S5S/pdb]. The source data underlying Figs. 1, 4, 5, 7A, B, and Supplementary Figs. 1, 5, and 8 are provided as a Source Data file. Source data are provided with this paper.

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

## Acknowledgements

Data were collected at the LCLS, SLAC National Accelerator Laboratory (proposal no. L10057), supported by the DOE Office of Science, OBES under contract no. DE-AC02-76SF00515. SFX data processing was performed in part at the National Energy Research Scientific Computing Center, supported by the DOE Office of Science, contract no. DEAC02-05CH11231. The Rayonix detector used at LCLS was supported by the NIH grant S10 OD023453. Data processing was supported by the US DIALS National Resource (R24GM154040). Experiments at the LCLS were supported by the NIH grant P41GM139687, and additional assistance and technical support from Dr Vandana Tiwari and Dr Humberto Sanchez is acknowledged. The ID29 beamline at the European Synchrotron Radiation Facility (ESRF) is acknowledged for early sample testing using a tape-drive delivery setup under BAG number MX2438. L.J.W. was supported by a joint studentship award (STU0436) from Diamond Light Source and the University of Essex. J.A.R.W. and M.A.H. acknowledge support from the BBSRC (BB/W001950/1). A.M.O. is supported by the Wellcome Trust (210734/Z/18/Z) and a Royal Society Wolfson Fellowship (RSWF\R2\182017). We acknowledge the XFEL hub at Diamond Light Source for travel assistance to LCLS to perform experiments. Support from the Romanian Ministry for Education and Research and the European Union—NextGenerationEU for the Romanian Government, under the National Recovery and Resilience Plan for Romania and the European Fund of Regional Development through the Competitiveness Operational Programme 2014-2020 (projects PNRR-III-C9-2023-I8-CF76, POC/398/1/1/124155, 390005/23.10.2024-INSPIRE-II, 235/2020-CLOUDUT) is gratefully acknowledged. U.R. acknowledges support of grants from the Swedish research council (projects 2020-06176 and 2022-04978). The computations were enabled by resources provided by LUNARC, the Centre for Scientific and Technical Computing at Lund University. Support from the U.S. Department of Energy, Office of Science (OS), Office of Basic Energy Sciences (BES), Chemical Sciences, Geosciences, and Biosciences Division, contract no. DE-AC02-05CH11231 (J.Y., V.K.Y., and J.F.K.) and by the National Institutes of Health (NIH) Grants GM149528 (V.K.Y.), GM110501 (J.Y.), GM126289 (J.F.K.), is acknowledged.

## Author contributions

L.J.W., M.A.H., R.L.S-D., J.A.R.W.: conceptualization. L.J.W., J.J.A.G.K., A.M.V.B., Maria L., K.J.M.L., K.C., M.A.D., P.S.S., H.M., Marina L., M.T.W., P.A., J.S.W., L.G., S.D., S.M., U.R., A.M.O., J.F.K., R.L.S.-D., and J.A.R.W.: investigation. A.J.T., A.S.B., T.Z., K.C., and M.A.D.: formal analysis. J.Y., V.K.Y., J.F.K., A.M.O., K.J.M.L., and U.R.: methodology. L.J.W., M.T.W., R.L.S-D., and J.A.R.W.: writing—original draft. J.F.K., U.R., M.A.H., K.C., L.J.W., M.T.W., R.L.S-D., and J.A.R.W.: writing—review and editing.

## Competing interests

The authors declare no competing interests.
