## [Transparent Peer Review file · Nature Communications]

Can ferric-oxyl excited states explain elongated iron-oxygen bonds in heme peroxidase catalytic intermediates?

Corresponding Author: Professor Jonathan Worrall

Version 0:

Reviewer comments:

Reviewer #1

(Remarks to the Author)

This extensive paper uses time-resolved serial femtosecond X-ray crystallisation on reactive Compound I and Compound II intermediates of dye-decolourising peroxidase. The drop-on-tape methodology that has been used in the paper has never (to my knowledge) been applied to a peroxidase system, and can reveal time-resolved information on reactive species that has not been previously accessible. It is coupled in this paper with parallel X-ray emission spectra, alongside DFT and QM/MM modelling. The authors present evidence that the iron-oxygen bond lengths in their structures are more consistent when considered as singly bond species, rather than with the expected double bond formulation. There is a very topical, and at times controversial, debate on-going about the nature of these heme species, and in this sense the paper is very timely.

What is nice about the paper is that it offers something quite new to this on-going debate – a new way of thinking about these species from a chemical perspective. In this sense, I found the paper to be interesting and highly insightful, and it definitely moves the debate (which to some extent had stalled for lack of new methods/data) forward.

For these reasons, I was very positive about the paper. I would be happy to see it published, and it definitely speaks to a wide audience, subject to some clarifications as below. A number of these clarifications are quite minor.

The colours of the CI and CII spectra in Figures 1 and 5 are different – keep consistent? Green for CI, as in Fig 5, seems sensible.

On p9, the authors look at spectra of microcrystals. The maxima in Fig 5 match those in Fig 1 at 418 nm, but the peak height is much lower for the 418 nm band in the crystals. Do the authors have an explanation for that?

The bonds lengths in Table 1 have errors that, for the most part, mean that the Fe-O distance are outside of the expected range for Fe=O. The exception of the $t > 20$ min chain B structure (1.81 Å with a 0.15 error) – do the authors consider this is within range of the ca 1.65 Å tolerance mentioned in the Introduction? The other possible (??) exception is the $t > 20$ min chain A structure (1.83 Å with a 0.14 error). Perhaps a comment on this would be helpful?

The key to the conclusions presented in the paper is the computational data on pages 9-12. I found the presentation of this part of the paper to be written for a highly specialised reader, with few concessions made for a general audience such as in Nature Comm. It would be a pity if this part of the paper is not accessible to readers, and I have a few comments/suggestions which would help a general readership.

- Lines 244-247 and Scheme are difficult to access. A much better legend to Scheme 1 is needed, with labelling of the bonding orbitals that links clearly to the text. It is taken as read by the authors that readers will understand Scheme and its origin (mentioned on line 244), but a better explanation of the bonding diagrams would be very helpful (e.g explaining where the p-orbitals come from, which t_{2g} orbitals they are presenting, what X_n means etc). Also, I did have to stare at Scheme 1 quite hard to capture the relationship/movements of the red electrons (which are correct, but not immediately obvious).

- It is not completely clear on line 249 which orbitals are being referred to as non-bonding – please name the xy and x^2-y^2 again for complete clarity. Ditto line 250-251 – the description of the pi bonding orbitals (which ones?). Please tighten up the explanations. Equally, lines 266-267 please state clearly which antibonding orbitals are being referred to.

- Please explain for the sake of non-specialist why the bond order goes down and hence why the bond length would increase (obvious to some, but not all).

- Lines 260-265 – why does the S=3 species in Table 2 not give a long bond if a σ^* orbital is occupied?

- Lines 264-265 – apologies if I missed it in the time I had, but the authors rule out higher spin states than S=1 but I didn't see anywhere in the abstract or conclusions what their final statement on the spin state was. If not made clear, then Scheme 1 is slightly distracting if it does present their final views and could be misconstrued. Lines 303-305 relate, and this might be a good place to re-state the final conclusions on the bonding interactions (MO diagram).

- Lines 269-283. The presentation of the MOs in Figure S4 is not easy to grasp. S4 needs a much better legend. To re-visit

this section and ensure it is understandable by non-specialists.

- Line 288 – it is not clear what the first excited state refers to (root 1). Ditto line 318, which mentioned the second excited state (root 2?). I might have missed it, but it was not clear to me how root 1 and root 2 related to the metal-oxyl descriptions in the main text.
- Line 313 – is the Fe³⁺-O- species the same or different from the metal oxyl (radical?).
- Line 292 – not clear what is meant in the figure showing excitation to a pi* orbital.
- Line 293 – the pi orbitals referred to are not labelled in 6A. Better legends for all the computational figures would help.
- Lines 355-366 considers Scheme 1 in relation to Figure 6. But navigating these figures (or connecting them together) is not straightforward. The increased spin density mentioned on line 361 isn't obvious to a non-specialist. Would a simple binding diagram help (somewhere?) Just showing the ferryl/ferric-oxyl simple bonding scheme – that seems not to have been included at all and yet is at the heart of it all.

Reviewer #2

(Remarks to the Author)

This paper is an important contribution to the long-standing controversy over the nature of the “Compound II” intermediates in heme peroxidases. Put simply, the questions are 1) What is the bond order of the iron-oxygen link? 2) Is the oxygen protonated? The use of the iron-oxygen distance as a proxy for bond order has dominated this debate. The authors successfully and critically outline the evidence that has been collected over decades. The authors propose, on basis of their studies of a slow reacting mutant of a dye-decolourising heme peroxidase (DyP) using XFEL, XES and optical spectroscopy along with computational simulations, propose that for Compound II, (which would be assumed to be Fe(IV)-OH species on the basis of the observed Fe-O distance) is rather an excited state better described as Fe(III)-O. The Fe(IV)-OH model that has been previously been suggested to explain the long Fe-O distance has been questioned on the basis of the pKa with a histidine proximal ligand. This “long” distance (ca. 1.85Å) is also seen in all other heme peroxidase and myoglobin Compound II structures apart from Cytochrome c Peroxidase. The spread of observed values has been attributed to the different active site environments. The only direct observation of the ferryl iron being ligated by OH has been in the neutron structure of APX Compound II. This would be inconsistent with the proposal here, should the Fe(III)-O model be universal. The observation here that the distance does not increase between Compound I and Compound II in DyP is inconsistent with the observations reported for other heme peroxidases, where Compound I is shorter (and interpreted as Fe(IV)=O). The observed average Fe-O distance of ~1.83 Å for both Compounds I and II and the suggestion of Fe(III)-oxyl character supported by TD-DFT is indeed intriguing. The structures presented here are over a series of time points, with the two different heme sites in the enzyme interpreted as evolving Compound I and Compound II at different rates from the initial ferric-water structure, although only two states are reported. This means that other evidence has had to be provided to show that the structures of both Compound I and II are determined and the authors are not seeing a mixtures of states.

Given the challenges of accurately refining light atom positions (especially when adjacent to more electron-dense atoms) and the crucial differences in interpretation of cry small changes, how confident can the authors be in distinguishing between an Fe=O distance of ~1.65–1.75 Å and the reported ~1.83 Å? The sigmas presented are for the Fe-O distances, it would be interesting to see the ESUs for the individual atoms. I have been unable to access the DPI estimator site cited, to verify that it gives sigmas on distances rather than positions (as I thought). Further uncertainty in the structures is added as there is local disorder. Anisotropic B-factors do not seem to have been used, this would be normal for the higher resolution structures. It would be informative to know the restraints used during refinement. It would also be helpful to report on the uncertainty of previously reported structures when the results are being discussed.

The spectral assignments for Compound I in both WT and Y389F require further clarification. In the WT enzyme, the earliest spectrum is described as red-shifted relative to canonical Compound I, raising the possibility that the species could represent a mixture or a protein-tuned form of C1. Conversely, in the Y389F variant the 401 nm Soret maximum is presented as a “near fully populated Compound I,” yet this differs from many reported Compound I spectra in related heme enzymes. How can the authors be certain that both the WT red-shifted intermediate and the 401 nm Y389F species correspond to bona fide Compound I rather than alternative oxidised states (e.g., admixtures of C1/II or delocalised radical species)? A clearer explanation of Figure S1 would help.

On P4, line 82, sources reference 10 for the shorter distance in Compound I in CcP, but this paper only reports an intermediate distance (1.76 Å) for Compound II.

I found the account of the DFT and QM/MM calculations challenging, although they seem to provide a sound justification for the question the authors have set in the title. To this reviewer, the argument that observed “long” CII Fe-O distances can be reconciled with the chemical arguments about the pKa are important and should be published. However, the observation that Compound I has an equally “long” distance seems at variance with so much of the literature that the (subjectively) convoluted arguments for this are less convincing.

Version 1:

Reviewer comments:

Reviewer #1

(Remarks to the Author)

The authors have made extensive revisions to the paper, and have substantially modified the description of the computation data. The new MO diagrams (Figure 6 and 7C) are much clearer and the text describing this is much clearer throughout. I am happy to see this paper now published.

Reviewer #2

(Remarks to the Author)

The revisions (more clarifications) have greatly improved the accessibility of the paper. The authors have satisfactorily resolved the issues raised in the first round of review.

It would be nice if the updated hyperlink to the DPI server could be included (e.g. in Table S2) to save those of us who went through the cited references and failed to find it.

It would also be appreciated if the dark blue arrows in figure S7 could be adjusted to improve the contrast from the black bond lines.

Minor edits needed from the changes I notices are: line 84 CCP should have the second C lower case (and this should be in italics throughout)

line 94 I think that the word "all" that has been inserted should have been after "yields" rather than before.

I recommend publication, I gave the justification for this on the first version

Reviewer 1

The colours of the CI And CII spectra in Figures 1 and 5 are different – keep consistent? Green for CI, as in Fig 5, seems sensible.

As requested, we have made the CI (Compound I) spectrum in both Figures 1 and 5 green. To make it stand out in Figure 1, we have also increased the thickness of the line.

On p9, the authors look at spectra of microcrystals. The maxima in Fig 5 match those in Fig 1 at 418 nm, but the peak height is much lower for the 418 nm band in the crystals. Do the authors have an explanation for that?

We subjected DtpAa microcrystal slurries loaded into a cuvette to conventional optical spectroscopy. The purpose was to identify the time evolution of the ferryl intermediates by using peak maxima. With hindsight we should have implemented some continuous agitation of the slurry as we believe that some crystal settling would have occurred during the 20 min experiment following H₂O₂ addition, which would affect the absolute absorbance (as questioned by the reviewer) but not of the course peak position of the species.

The bonds lengths in Table 1 have errors that, for the most part, mean that the Fe-O distance are outside of the expected range for Fe=O. The exception of the t>20 min chain B structure (1.81 Å with a 0.15 error) – do the authors consider this is within range of the ca 1.65 Å tolerance mentioned in the Introduction? The other possible (??) exception is the t>20 min chain A structure (1.83 Å with a 0.14 error). Perhaps a comment on this would be helpful?

We have extensively addressed a similar point raised by reviewer 2. We understand why this concern is raised however, we point out that our error estimates are essentially an upper bound. As the resolution of the t>20 min structure was lower than that of the earlier time-points (1.5 versus 1.9 Å), the error is correspondingly larger. The Fe-O distance after refinement was essentially invariable across the time series giving confidence that they are true values. We have added a comment in the text (page 7) about the resolution of the structure being related to the uncertainty of the atomic positions.

The key to the conclusions presented in the paper is the computational data on pages 9-12. I found the presentation of this part of the paper to be written for a highly specialised reader, with few concessions made for a general audience such as in Nature Comm. It would be a pity if this part of the paper is not accessible to readers, and I have a few comments/suggestions which would help a general readership.

Yes, we accept this criticism and have endeavoured to address the comments with additional text and a new figure.

- Lines 244-247 and Scheme are difficult to access. A much better legend to Scheme 1 is needed, with labelling of the bonding orbitals that links clearly to the text. It is taken as read by the authors that readers will understand Scheme and its origin (mentioned on line 244), but a better explanation of the bonding diagrams would be very helpful (e.g explaining where the p-orbitals come from, which t_{2g} orbitals they are presenting, what X_n- means etc). Also, I did have to stare at Scheme 1 quite hard to capture the relationship/movements of the red electrons (which are correct, but not immediately obvious).

We have converted the original Scheme 1 for a ferryl (Fe^{IV}=O) system into a figure (Figure 6) and given it an appropriate legend. In this new figure the orbitals are now labelled and drawn

and fully integrated into the main text. The red arrows are now only with their corresponding excited state with their movement between orbitals indicated. This now makes it easier for the reader to locate and capture their movement within excited states (as further discussed in the main text).

- It is not completely clear on line 249 which orbitals are being referred to as non-bonding – please name the xy and x^2-y^2 again for complete clarity. Ditto line 250-251 – the description of the π bonding orbitals (which ones?). Please tighten up the explanations. Equally, lines 266-267 please state clearly which antibonding orbitals are being referred to.

We have now made clear throughout the text (pages 9-11) which orbitals we are referring to, and which are bonding and anti-bonding and make continuous reference to Figure 6. The Fe-O orbitals are also now mentioned in the new legend to Figure 6 and in the updated Figure 7.

- Please explain for the sake of non-specialist why the bond order goes down and hence why the bond length would increase (obvious to some, but not all).

At the bottom of page 9 and top of page 10, we have inserted text that provides the explanation sought by the reviewer, with the new Figure 6 helping with our explanation.

- Lines 260-265 – why does the $S=3$ species in Table 2 not give a long bond if a σ^ orbital is occupied?*

The reviewer is referring to Table S2 (now S3 in revised version). In this Table, the non-heme ferryl system, has a longer bond, but for the ferryl heme system all electronic spin states maintain a short Fe-O bond (~ 1.6 Å) – suggesting that the second excitation would involve the porphyrin contributions. Text inserted on page 10 to reflect this more clearly.

- Lines 264-265 – apologies if I missed it in the time I had, but the authors rule out higher spin states than $S=1$ but I didn't see anywhere in the abstract or conclusions what their final statement on the spin state was. If not made clear, then Scheme 1 is slightly distracting if it does present their final views and could be misconstrued. Lines 303-305 relate, and this might be a good place to re-state the final conclusions on the bonding interactions (MO diagram).

Yes we are ruling out higher spin states, as explained now more clearly in the revised text and with the replacement of scheme 1 with Figure 6. We are proposing excitations that do not involve higher spin states such as those shown in Figure 6B and 7C. Hopefully these new figures make this clearer.

- Lines 269-283. The presentation of the MOs in Figure S4 is not easy to grasp. S4 needs a much better legend. To re-visit this section and ensure it is understandable by non-specialists.

We have edited the legend to Figure S4 and hopefully it is now clearer. In the main text where we refer to Figure S4, we have now mentioned the orbitals in relation to the new Figure 6 and have also signposted the reader to which molecular orbitals in Figure S4 we are referring to in the main text (i.e. red, blue or black box boxes).

- Line 288 – it is not clear what the first excited state refers to (root 1). Ditto line 318, which mentioned the second excited state (root 2?). I might have missed it, but it was not clear to me how root 1 and root 2 related to the metal-oxyl descriptions in the main text.

Root 1 and root 2 mentioned in Table 2 refer to the lowest-energy excited state and second lowest, respectively. We have edited Table 2 and made this clear elsewhere in the text.

- Line 313 – is the $\text{Fe}^{3+}\text{-O}^-$ species the same or different from the metal oxyl (radical?).

Yes, it is the same species. We have now made the nomenclature consistent throughout the text; ferric-oxyl ($\text{Fe}^{\text{III}}\text{-O}^-$).

- Line 292 – not clear what is meant in the figure showing excitation to a π^* orbital.

We have now clarified in the main text the orbitals involved (page 10; QM/MM section), and we have better described this transition in the legend to Figure S6, including a better presentation of the table.

- Line 293 – the π orbitals referred to are not labelled in 6A. Better legends for all the computational figures would help.

We have extensively edited the computational figure legends in the main text and the supplementary information. We hope the reviewer finds them clearer. With respect to the π orbitals, we have included these in the legend to Figure 7 (old Figure 6).

- Lines 355-366 considers Scheme 1 in relation to Figure 6. But navigating these figures (or connecting them together) is not straightforward. The increased spin density mentioned on line 361 isn't obvious to a non-specialist. Would a simple binding diagram help (somewhere?) Just showing the ferryl/ferric-oxyl simple bonding scheme – that seems not to have been included at all and yet is at the heart of it all.

The bonding scheme requested by the reviewer is now given as Figure 7C. We have edited the figure legend of Figure 7 (old Figure 6) to highlight the spin density change in Figure 7D to which we are referring to in the Discussion.

Reviewer 2

Given the challenges of accurately refining light atom positions (especially when adjacent to more electron-dense atoms) and the crucial differences in interpretation of cryo small changes, how confident can the authors be in distinguishing between an Fe=O distance of ~1.65–1.75 Å and the reported ~1.83 Å?

The DPI metric used here to estimate errors, was originally benchmarked against errors obtained from SHELXL full matrix inversion by Cruickshank (Cruickshank D. *Acta Crystallogr D* **55**, 583-601 (1999)) revealing good correspondence. In the three examples given (with resolutions between 0.94 and 1.70 Å) in the original DPI paper the error estimate obtained from the DPI was in fact somewhat larger than that obtained from SHELXL suggesting that it represents an upper bound. In addition, we note that we typically observe highly consistent bond length values for Fe-O distances between different crystals in our other reported heme peroxidase structures whether determined from serial crystallography or in structures determined using cryo-cooled crystals. For all the time-resolved structures reported in this manuscript, where the heme has reacted with H₂O₂ the refined bond-lengths showed little

variation. For these reasons we do indeed have confidence in defining the bond lengths for the heme intermediate species across the time series, although of course the error is as expected somewhat larger for the lower resolution structures.

The sigmas presented are for the Fe-O distances, it would be interesting to see the ESUs for the individual atoms. I have been unable to access the DPI estimator site cited, to verify that it gives sigmas on distances rather than positions (as I thought).

Note that the server for DPI calculation is now hosted here: <http://pdbi.nii.ac.in/~server/dpi/> The DPI server indeed gives sigmas for individual atomic positions. We have then used the method described by Helliwell and co-workers (Gurusaran, M., Shankar, M., Nagarajan, R., Helliwell, J. R. & Sekar, K. (2014). IUCrJ, 1, 74-81) to convert the individual error estimates for two atoms into an error of the distance between them. We have added a supplementary table (Table S2) to the manuscript giving individual coordinate DPI values alongside the uncertainties of the bond lengths for the Fe-O distances in our structures and the equation used to calculate this error (see supporting information).

Further uncertainty in the structures is added as there is local disorder. Anisotropic B-factors do not seem to have been used, this would be normal for the higher resolution structures.

Anisotropic B-factors are sometimes but not typically used at 1.5 Å resolution and on a case-by-case basis. The range from 1.3-1.5 Å has been called the grey zone for the decision of whether to use isotropic or anisotropic temperatures [e.g. Jaskolski, M.(2017) "Structure Refinement at Atomic Resolution" in Chapter 22 page 549 of *Alexander Wlodawer et al. (eds.), Protein Crystallography: Methods and Protocols, Methods in Molecular Biology, vol. 1607*]. In our study we did not do this and note that for consistency in how the datasets were handled at varying resolutions we would not want to refine certain structures anisotropically and certain ones isotropically.

It would be informative to know the restraints used during refinement.

We used standard restraint libraries for refinement in Phenix – and specifically we did not restrain the Fe-O or Fe-N(His) distances. We have added this text to page 18.

It would also be helpful to report on the uncertainty of previously reported structures when the results are being discussed.

Where we have quoted bond-lengths from other structures we have now reported the error either from the publication or by running the deposited PDB file through the DPI server (pages 4 and 14). We have not included an error for the neutron structure because of the low completeness of the data causing it to be rejected by the DPI server.

The spectral assignments for Compound I in both WT and Y389F require further clarification. In the WT enzyme, the earliest spectrum is described as red-shifted relative to canonical Compound I, raising the possibility that the species could represent a mixture or a protein-tuned form of CI.

For the WT enzyme, by employing static electronic absorption spectroscopy we are capturing a mixture of Compound I/II. We have edited the plot legend of Figure 1A to make this clear as well as the Figure legend – we have not coloured the first spectrum of the WT, green, as suggested by reviewer 1, as it does not represent pure Compound I but a mixture. Text has also been altered on page 5 to reflect this point.

Conversely, in the Y389F variant the 401 nm Soret maximum is presented as a “near fully populated Compound I,” yet this differs from many reported Compound I spectra in related heme enzymes.

For the variant, using static spectroscopy we capture in 60 seconds a much higher proportion of Compound I, and the peak maxima positions now correlate with previous Compound I spectra obtained from rapid mixing of ferric states with H₂O₂ using stopped-flow absorption spectroscopy of both A-type DyPs, and that of a B-type DyP which has a Compound I stable for > 1 h. We added references and edited the text on page 5.

How can the authors be certain that both the WT red-shifted intermediate and the 401 nm Y389F species correspond to bona fide Compound I rather than alternative oxidised states (e.g., admixtures of C1/II or delocalised radical species)? A clearer explanation of Figure S1 would help.

Figure 1 nicely demonstrates that Compound I lifetime is longer in the variant and is used to rationalise why we chose this variant for tr-SFX studies. To consolidate this observation rapid-mixing stopped-flow was used (Figure S1) which shows that Compound I in the variant is fully populated for the time points-measured in our tr-SFX experiments. We have edited the text on page 5, to make clear this point, and also clarified Figure S1 by indicating that the formation and decay of Compound I is represented on two x axis with different time scales.

On P4, line 82, sources reference 10 for the shorter distance in Compound I in CcP, but this paper only reports an intermediate distance (1.76 Å) for Compound II.

Apologies for this oversight. This has now been corrected and the text edited accordingly on page 4.

To this reviewer, the argument that observed “long” CII Fe-O distances can be reconciled with the chemical arguments about the pKa are important and should be published. However, the observation that Compound I has an equally “long” distance seems at variance with so much of the literature that the (subjectively) convoluted arguments for this are less convincing.

We have gone to great lengths to be certain we are capturing Compound I and Compound II in our tr-SFX and XES experiments. We are confident that the bond distances correspond to both Compound I and II in these experiments. The chemical argument we put forward is the same regardless of whether it is a Compound I or II species.

Reviewer 1

The authors have made extensive revisions to the paper, and have substantially modified the description of the computation data. The new MO diagrams (Figure 6 and 7C) are much clearer and the text describing this is much clearer throughout. I am happy to see this paper now published.

No changes requested. We thank the reviewer for helping us improve our manuscript.

Reviewer 2

The revisions (more clarifications) have greatly improved the accessibility of the paper. The authors have satisfactorily resolved the issues raised in the first round of review.

We thank the reviewer for helping us to improve our manuscript.

It would be nice if the updated hyperlink to the DPI server could be included (e.g. in Table S2) to save those of us who went through the cited references and failed to find it.

The working hyperlink link to the DPI server is now included in the legend to Table S2

It would also be appreciated if the dark blue arrows in figure S7 could be adjusted to improve the contrast from the black bond lines.

As requested, we have edited the figure and have remade the vector arrows in red.

Minor edits needed from the changes I notices are: line 84 CCP should have the second C lower case (and this should be in italics throughout)

Corrected.

line 94 I think that the word "all" that has been inserted should have been after "yields" rather than before.

Corrected.